# EFFICIENT EXPLORATION THROUGH BAYESIAN DEEP Q-NETWORKS

## ABSTRACT

We propose Bayesian Deep Q-Network (BDQN), a practical Thompson sampling based Reinforcement Learning (RL) Algorithm. Thompson sampling allows for targeted exploration in high dimensions through posterior sampling, but is usually computationally expensive. We address this limitation by introducing uncertainty only at the output layer of the network through a Bayesian Linear Regression (BLR) model, which can be trained with fast closed-form updates and its samples can be drawn efficiently through the Gaussian distribution. We apply our method to a wide range of Atari Arcade Learning Environments. Since BDQN carries out more efficient exploration, it is able to reach higher rewards substantially faster than a key baseline, DDQN.

## 1 INTRODUCTION

Designing algorithms that achieve an optimal trade-off between exploration and exploitation is one of the primary goal of reinforcement learning (RL). However, targeted exploration in high dimensional spaces is a challenging problem in RL. Recent advances in deep RL mostly deploy simple exploration strategies such as $\varepsilon$-greedy, where the agent chooses the optimistic action, the action with highest promising return, with probability $(1 - \varepsilon)$, otherwise, uniformly at random picks one of the available actions. Due to this uniform sampling, the $\varepsilon$-greedy method scales poorly with the dimensionality of state and action spaces. Recent work has considered scaling exploration strategies to large domains (Bellemare et al., 2016; Tang et al., 2016). Several of these papers have focused on employing optimism-under-uncertainty approaches, which essentially rely on computing confidence bounds over different actions, and acting optimistically with respect to that uncertainty.

An alternative to optimism-under-uncertainty (Brafman & Tennenholtz, 2003) is Thompson Sampling (TS) (Thompson, 1933), one of the oldest heuristics for multi arm bandits. TS is a Bayesian approach where one starts with a prior distribution over the belief and compute the posterior beliefs based on the collected data through the interaction with the environment and then maximizes the expected return under the sampled belief. The TS based posterior sampling can provide more targeted exploration since it can trade off uncertainty with the expected return of actions. In contrast, the $\varepsilon$-greedy strategy is indifferent to uncertainty of the actions and the expected rewards of sub-optimistic ones (set of actions excluding the optimistic action).

There has been relatively little work on scaling Thompson Sampling to large state spaces. The primary difficulty in implementing Thompson sampling is the difficulty of sampling from general posterior distributions. Prior efforts in this space have generally required extremely expensive computations (e.g. (Ghavamzadeh et al., 2015; Strens, 2000))

We derive a practical Thompson sampling framework, termed as Bayesian deep Q-networks (BDQN), where we approximate the posterior distribution on the set of Q-functions and sample from this approximated posterior. BDQN is computationally efficient since it incorporates uncertainty only at the output layer, in the form of a Bayesian linear regression model. Due to linearity and by choosing a Gaussian prior, we derive a closed-form analytical update to the approximated posterior distribution over Q functions. We can also draw samples efficiently from the Gaussian distribution. As addressed in Mnih et al. (2015), one of the major benefits of function approximation methods in deep RL is

that the estimation of the Q-value, given a state-action pair, can generalize well to other state-action pairs, even if they are visited rarely. We expect this to hold in BDQN as well, but additionally, we also expect the uncertainty of state-action pairs to generalize well.

We test BDQN on a wide range of Arcade Learning Environment Atari games (Bellemare et al., 2013; Brockman et al., 2016) against a strong baseline DDQN (Van Hasselt et al., 2016). Aside from simplicity and popularity of DDQN, BDQN and DDQN share the same architecture, and follow same target objective. These are the main reasons we choose DDQN for our comparisons.

In table. 1 we see significant gains for BDQN over DDQN. BDQN is able to learn significantly faster and reach higher returns due to more efficient exploration. The evidence of this is further seen from the fact that we are able to train BDQN with much higher learning rates compared to DDQN. This suggests that BDQN is able to learn faster and reach better scores.

These promising results suggest that BDQN can further benefit from additional modifications that were done to DQN, e.g. Prioritized Experience Replay (Schaul et al., 2015), Dueling approach (Wang et al., 2015), A3C (Mnih et al., 2016), safe exploration (Lipton et al., 2016a), and etc. This is because BDQN only changes that exploration strategy of DQN, and can easily accommodate additional improvements to DQN.

| Game | $\frac{\text{BDQN}}{\text{DDQN}}$ | $\frac{\text{BDQN}}{\text{DDQN}^{\dagger}}$ | $\frac{\text{BDQN}}{\text{HUMAN}}$ | Step |
|---|---|---|---|---|
| Amidar | 558% | 788% | 325% | 100M |
| Alien | 103% | 103% | 43% | 100M |
| Assault | 396% | 176% | 589% | 100M |
| Asteroids | 2517% | 1516% | 108% | 100M |
| Asterix | 531% | 385% | 687% | 100M |
| BeamRider | 207% | 114% | 150% | 70M |
| BattleZone | 281% | 253% | 172% | 50M |
| Atlantis | 80604% | 49413% | 11172% | 40M |
| DemonAttack | 292% | 114% | 326% | 40M |
| Centipede | 114% | 178% | 61% | 40M |
| BankHeist | 211% | 100% | 100% | 40M |
| CrazyClimber | 148%k | 122% | 350% | 40M |
| ChopperCommand | 14500% | 1576% | 732% | 40M |
| Enduro | 295% | 350% | 361% | 30M |
| Pong | 112% | 100% | 226% | 5M |

Table 1: The first column presents the score ratio after last column steps. The second column is the score ratio of BDQN after the number of steps in the last column compared to the score of DDQN$^{\dagger}$, the reported scores of DDQN in Van Hasselt et al. (2016) after running for 200M samples during evaluation time, and the third column is with respect to Human score reported at Mnih et al. (2015).

## 2  RELATED WORK

The complexity of the exploration-exploitation trade-off has been vastly investigated in RL literature (Kearns & Singh, 2002; Brafman & Tennenholtz, 2003; Strehl et al., 2009; Bartlett & Tewari, 2009; Azar et al., 2017; Dann et al., 2017). Auer (2002) addresses this question for multi-armed bandit problems where regret guarantees are provided. Jaksch et al. (2010) investigates the regret analyses in MDPs where exploration happens through the optimal policy of optimistic model, a well-known Optimism in Face of Uncertainty (OFU) principle where a high probability regret upper bound is guaranteed. Azizzadenesheli et al. (2016a) deploys OFU in order to propose the high probability regret upper bound for Partially Observable MDPs (POMDPs) and finally, Bartók et al. (2014) tackles a general case of partial monitoring games and provides minimax regret guarantee which is polynomial in certain dimensions of the problem.

In multi arm bandit, there is compelling empirical evidence that Thompson Sampling can provide better results than optimism-under-uncertainty approaches (Chapelle & Li, 2011), while the state of the art performance bounds are preserved (Russo & Van Roy, 2014; Agrawal & Goyal, 2012; Gopalan & Mannor, 2015; Abeille & Lazaric, 2017). A natural adaptation of this algorithm to RL, posterior sampling RL (PSRL), first proposed by Strens (2000) also shown to have good frequentist and Bayesian performance guarantees (Osband et al., 2013; Gopalan & Mannor, 2015; Agrawal & Jia, 2017; Abbasi-Yadkori & Szepesvári, 2015; Ouyang et al., 2017; Theocharous et al., 2017; Russo et al., 2017). Even though the theoretical RL addresses the exploration and exploitation trade-offs, these problems are still prominent in empirical reinforcement learning research (Mnih et al., 2015; Jiang et al., 2016; Abel et al., 2016; Azizzadenesheli et al., 2016b). On the empirical side, the recent success in the video games has sparked a flurry of research interest. Following the success of Deep RL on Atari games (Mnih et al., 2015) and the board game Go (Silver et al., 2017), many researchers have begun exploring practical applications of deep reinforcement learning (DRL). Some investigated applications include, robotics (Levine et al., 2016), energy management (Night, 2016), and self-driving cars (Shalev-Shwartz et al., 2016). Among the mentioned literature, the prominent exploration strategy for Deep RL agent has been $\varepsilon$-greedy.

Inevitably for PSRL, the act of posterior sampling for policy or value is computationally intractable with large systems, so PSRL can not be easily leveraged to high dimensional problems. To remedy these failings Osband et al. (2017) consider the use of randomized value functions to approximate posterior samples for the value function in a computationally efficient manner. They show that with a suitable linear value function approximation, using the approximated Bayesian linear regression for randomized least-squares value iteration method can remain statistically efficient (Osband et al., 2014) but still is not scalable to large scale RL with deep neural networks.

To combat these shortcomings, Osband et al. (2016) suggests a bootstrapped-ensemble approach that trains several models in parallel to approximate uncertainty. Other works suggest using a variational approximation to the Q-networks (Lipton et al., 2016b) or noisy network (Fortunato et al., 2017). However, most of these approaches significantly increase the computational cost of DQN and neither approaches produced much beyond modest gains on Atari games. Applying Bayesian regression in last layer of neural network was investigated in Snoek et al. (2015) for object recognition, image caption generation, and etc. where a significant advantage has been provided.

In this work we present another alternative approach that extends randomized least-squares value iteration method (Osband et al., 2014) to deep neural networks: we approximate the posterior by a Bayesian linear regression only on the last layer of the neural network. This approach has several benefits, e.g. simplicity, robustness, targeted exploration, and most importantly, we find that this method is much more effective than any of these predecessors in terms of sample complexity and final performance.

Concurrently, O'Donoghue et al. (2017) studies how to construct the frequentist confidence of the regression on the feature space of the neural network for RL problems by learning the uncertainties through a shallow network while Levine et al. (2017), similar to BDQN, suggests running linear regression on the representation layer of the deep network in order to learn the state-action value. Drop-out, as another randomized exploration method is proposed by Gal & Ghahramani (2016) but Osband et al. (2016) argues about its estimated uncertainty and hardness in driving a suitable exploitation of it. It is worth noting that most of proposed methods are based on $\varepsilon$-exploration.

## 3 THOMPSON SAMPLING VS $\varepsilon-$GREEDY

In this section, we enumerate a few benefits of TS over $\varepsilon$-greedy strategies. We show how TS strategies exploit the uncertainties and expected returns to design a randomized exploration while $\varepsilon-$greedy strategies disregard all these useful information for the exploration.

Fig. 1*(a)* expresses the agent's estimated values and uncertainties for the available actions at a given state $x$. While $\varepsilon-$greedy strategy mostly focus on the action 1, the optimistic action, the TS based strategy randomizes, mostly, over actions 1 through 4, utilizing their approximated returns and uncertainties, and with low frequency tries actions 5, 6. Not only the $\varepsilon-$greedy strategy explores actions 5 and 6, where the RL agent is almost sure about low return of these actions, as frequent as other sub-optimal actions, but also spends the network capacity to accurately estimate their values.

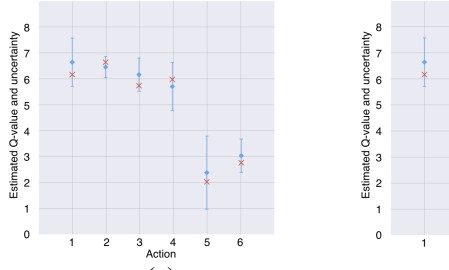 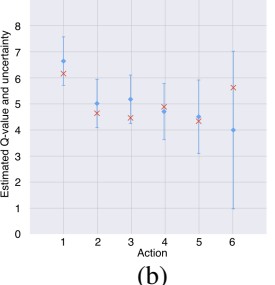 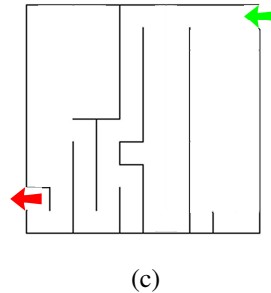

(a)  (b)  (c)

Figure 1: A cartoon representation of TS vs $\varepsilon$-greedy. The red crosses are the true Q-values, the diamonds are the mean of estimated Q-values with blue intervals as uncertainties (e.g. $c \cdot variance$)

A commonly used technique in deep RL is a moving window of replay buffer to store the recent experiences. The TS based agent, after a few tries of actions 5 and 6 builds a belief in low return of these actions given the current target values. Since the replay buffer is bounded moving window, lack of samples of these actions pushes the posterior belief of these actions to the prior, over time, and the agent tries them again in order to update its belief Fig. 1(b). In general, TS based strategy advances the exploration-exploitation trade-off by making trade-off between the expected returns and the uncertainties, while $\varepsilon-$greedy strategy ignores all of this information.

Another superiority of TS over $\varepsilon$-greedy can be described using Fig. 1(c). Consider an episodic maze-inspired deterministic game, with episode length $H$ of shortest pass from the start to the destination. The agent is placed to the start point at the beginning of each episode where the goal state is to reach the destination and receive a reward of 1 otherwise reward is 0. Consider an agent, which is given a hypothesis set of Q-functions where the true Q-function is within the set and is the most optimistic function in the set. In this situation, TS randomizes over the Q-functions with high promising returns and relatively high uncertainty, including the true Q-function. When it picks the true Q-function, it increases the posterior probability of this Q-function because it matches the likelihood. When TS chooses other functions, they predict deterministically wrong values and the posterior update of those functions set to zero, therefore, the agent will not choose them again, i.e. TS finds the true Q-function very fast. For $\varepsilon$-greedy agent, even though it chooses the true function at the beginning (it is the optimistic one), at each time step, it randomizes its action with the probability $\varepsilon$. Therefore, it takes exponentially many trials in order to get to the target in this game.

## 4  PRELIMINARIES

An infinite horizon $\gamma$-discounted MDP $M$ is a tuple $\langle \mathcal{X}, \mathcal{A}, T, R \rangle$, with state space $\mathcal{X}$, action space $\mathcal{A}$, and the transition kernel $T$, accompanied with reward function of $R$ where $0 < \gamma \leq 1$. At each time step $t$, the environment is at a state $x_t$, called current state, where the agent needs to make a decision $a_t$ under its policy. Given the current state and action, the environment stochastically proceed to a successor state $x_{t+1}$ under probability distribution $T(X_{t+1}|x_t, a_t) := \mathbb{P}(X_{t+1}|x_t, a_t)$ and provides a stochastic reward $r_t$ with mean of $\mathbb{E}[r|x = x_t, a = a_t] = R(x_t, a_t)$. The agent objective is to optimize the overall expected discounted reward over its policy $\pi := \mathcal{X} \to \mathcal{A}$, a stochastic mapping from states to actions, $\pi(a|x) := \mathbb{P}(a|x)$.

$$\eta^* = \eta(\pi^*) = \max_\pi \eta(\pi) = \max_\pi \lim_{N \to \infty} \mathbb{E}_\pi \left[ \sum_{t=0}^{N} \gamma^t r_t \right] \tag{1}$$

The expectation in Eq. 1 is with respect to the randomness in the distribution of initial state, transition probabilities, stochastic rewards, and policy, under stationary distribution, where $\eta^*, \pi^*$ are optimal return and optimal policy, respectively. Let $Q_\pi(x, a)$ denote the average discounted reward under policy $\pi$ starting off from state $x$ and taking action $a$ in the first place.

$$Q_\pi(x, a) := \lim_{N \to \infty} \mathbb{E}_\pi \left[ \sum_{t=0}^{N} \gamma^t r_t | x_0 = x, a_0 = a \right]$$

For a given policy $\pi$ and Markovian assumption of the model, we can rewrite the equation for the Q functions as follows:

$$Q_\pi(x, a) = R(x, a) + \gamma \sum_{x', a'} T(x'|x, a)\pi(a'|x')Q_\pi(x', a') \tag{2}$$

To find the optimal policy, one can solve the Linear Programing problem in Eq. 1 or follow the corresponding Bellman equation Eq. 2 where both of optimization methods turn to

$$Q^*(x, a) = R(x, a) + \gamma \sum_{x'} T(x'|x, a) \max_{a'} Q^*(x', a')$$

where $Q^*(x, a) = Q_{\pi^*}(x, a)$ and the optimal policy is a deterministic mapping from state to actions in $\mathcal{A}$, i.e. $x \to arg\max_a Q^*(x, a)$. In RL, we do not know the transition kernel and the reward function in advance, therefore, we can not solve the posed Bellman equation directly. In order to tackle this problem, (Lagoudakis & Parr, 2003; Antos et al., 2008) studies the property of minimizing the Bellman residual of a given Q-function

$$\mathcal{L}(Q) = \mathbb{E}_\pi \left[ (Q(x, a) - r - \gamma Q(x', a'))^2 \right] \tag{3}$$

Where the tuple $(x, a, r, x', a')$ consists of consecutive samples under behavioral policy $\pi$. Furthermore, (Mnih et al., 2015) carries the same idea, and introduce Deep Q-Network (DQN) where the Q-functions are parameterized by a DNN. To improve the quality of Q function, they use back propagation on loss $\mathcal{L}(Q)$ using TD update (Sutton & Barto, 1998). In the following we describe the setting used in DDQN. In order to reduce the bias of the estimator, they introduce target network $Q^{target}$ and target value $y = r + \gamma Q^{target}(x', \hat{a})$ where $\hat{a} = \arg\max_{a'} Q(x', a')$ with a new loss $\mathcal{L}(Q, Q^{target})$

$$\mathcal{L}(Q, Q^{target}) = \mathbb{E}_\pi \left[ (Q(x, a) - y)^2 \right] \tag{4}$$

Minimizing this regression loss, and respectably its estimation $\widehat{\mathcal{L}}(Q, Q^{target})$, matches the $Q$ to the target $y$. Once in a while, the algorithm sets $Q^{target}$ network to $Q$ network, peruses the regression with the new target value, and provides an biased estimator of the target.

## 5 BAYESIAN DEEP Q-NETWORKS

We propose a Bayesian method to approximate the $Q$-function and match it to the target value. We utilize the DQN architecture, remove its last layer, and build a Bayesian linear regression (BLR) (Rasmussen & Williams, 2006) on the feature representation layer, $\phi_\theta(x) \in \mathbb{R}^d$, parametrized by $\theta$. We deploy BLR to efficiently approximate the distribution over the Q-values where the uncertainty over the values is captured. A common assumption in DNN is that the feature representation is suitable for linear classification or regression (same assumption in DQN).

The Q-functions can be approximated as a linear combination of features, i.e. for a given pair of state-action, $Q(x, a) = \phi_\theta(x)^\top w_a$. Therefore, by deploying BLR, we can approximate the generative model of the Q-function using its corresponding target value: $y = r + \gamma \phi^{target} w^{target}_{\hat{a}}$, where $\phi_\theta^{target}(x) \in \mathbb{R}^d$ denotes the feature representation of target network, for any $(x, a)$ as follows

$$y = Q(x, a) + \epsilon = w_a^\top \phi(x) + \epsilon, \quad \forall x \in \mathcal{X}, a \in \mathcal{A}$$

where $\epsilon \sim \mathcal{N}(0, \sigma_\epsilon^2)$ is an iid noise. Furthermore, we consider $w_a \in \mathbb{R}^d$ for $a \in \mathcal{A}$ are drawn approximately from a Gaussian prior $\mathcal{N}(0, \sigma^2)$. Therefore, $y|x, a, w_a \sim \mathcal{N}(\phi(x)^\top w_a, \sigma_\epsilon^2)$. Moreover, the distribution of the target value $y$ is $\mathbb{P}(y|a) = \int_{w_a} \mathbb{P}(y|w_a)\mathbb{P}(w_a) dw_a$. Given a dataset $\mathcal{D} = \{x_\tau, a_\tau, y_\tau\}_{\tau=1}^D$, we construct $|\mathcal{A}|$ disjoint datasets for each action, $\mathcal{D}_a$, where $\mathcal{D} = \cup_{a \in \mathcal{A}} \mathcal{D}_a$ and $\mathcal{D}_a$ is a set of tuples $x_\tau, a_\tau, y_\tau$ with the action $a_\tau = a$ and size $D_a$. We are interested in $\mathbb{P}(w_a|\mathcal{D}_a)$ and $\mathbb{P}(Q(x, a)|\mathcal{D}_a), \forall x \in \mathcal{X}$. We construct a matrix $\Phi_a \in \mathbb{R}^{d \times D_a}$, a concatenation of feature column vectors $\{\phi(x_i)\}_{i=1}^{D_a}$, and $\mathbf{y}_a \in \mathbb{R}^{D_a}$, a concatenation of target values in set $\mathcal{D}_a$. Therefore the posterior distribution is as follows:

$$w_a \sim \mathcal{N}\left(\frac{1}{\sigma_\epsilon^2}\Xi_a\Phi_a\mathbf{y}_a, \Xi_a\right), \;\; where \;\; \Xi_a = \left(\frac{1}{\sigma_\epsilon^2}\Phi_a\Phi_a^\top + \frac{1}{\sigma^2}\mathbb{1}\right)^{-1} \tag{5}$$

---

**Algorithm 1** BDQN

1: Initialize parameter sets $\theta$, $\theta^{target}$, $W$, $W^{target}$, and $Cov$ using a normal distribution.
2: Initialize replay buffer and set counter $= 0$
3: **for** episode = 1 to inf **do**
4:     Initialize $x_1$ to the initial state of the environment
5:     **for** $t =$ to the end of episode **do**
6:         **if** count mod $T^{sample} = 0$ **then**
7:             sample $W \sim \mathcal{N}(W^{target}, Cov)$
8:         **end if**
9:         Select action $a_t = argmax_{a'} \left[W^\top \phi_\theta(x_t)\right]_{a'}$
10:        Execute action $a_t$ in environment, observing reward $r_t$ and successor state $x_{t+1}$
11:        Store transition $(x_t, a_t, r_t, x_{t+1})$ in replay buffer
12:        Sample a random minibatch of transitions $(x_\tau, a_\tau, r_\tau, x_{\tau+1})$ from replay buffer
13:        $y_\tau \leftarrow \begin{cases} r_\tau, & \text{for terminal } x_{\tau+1} \\ r_\tau + \left[W^{target \top} \phi_{\theta^{target}}(x_{\tau+1})\right]_{\hat{a}} \, ; \hat{a} = argmax_{a'} \left[W^\top \phi_\theta(x_{\tau+1})\right]_{a'} & \text{for non-terminal } x_{\tau+1} \end{cases}$
14:        $\theta \leftarrow \theta - \eta \cdot \nabla_\theta (y_\tau - \left[W^\top \phi_\theta(x_\tau)\right]_{a_\tau})^2$
15:        **if** count mod $T^{target} = 0$ **then**
16:            set $\theta^{target} \leftarrow \theta$
17:        **end if**
18:        **if** count mod $T^{Bayes \, target} = 0$ **then**
19:            Update $W^{target}$ and $Cov$
20:        **end if**
21:        count = count + 1
22:    **end for**
23: **end for**

---

$\mathbb{1} \in \mathbb{R}^d$ is a identity matrix, and $Q(x,a)|\mathcal{D}_a = w_a^\top \phi(x)$. Since the prior and likelihood are conjugate of each other we have the posterior distribution over the discounted return approximated as

$$\sum_{t=0}^{N} \gamma^t r_t | x_0 = x, a_0 = a, \mathcal{D}_a \sim \mathcal{N}\left(\frac{1}{\sigma_\epsilon^2} \phi(x)^\top \Xi_a \Phi_a \mathbf{y}_a, \phi(x)^\top \Xi_a \phi(x)\right) \tag{6}$$

The expression in Eqs. 5 gives the posterior distribution over weights $w_a$ and the function $Q$. As TS suggests, for the exploration, we exploit the expression in Eq. 5. For all the actions, we set $w_a^{target}$ as the mean of posterior distribution over $w_a$. For each action, we sample a wight vector $w_a$ in order to have samples of mean Q-value. Then we act optimally with respect to the sampled means

$$a_{\text{TS}} = \arg\max_a w_a^\top \phi_\theta(x). \tag{7}$$

Let $W = \{w_a\}_{a=1}^{|\mathcal{A}|}$, respectively $W^{target} = \{w_a^{target}\}_{a=1}^{|\mathcal{A}|}$, and $Cov = \{\Xi_a\}_{a=1}^{|\mathcal{A}|}$. In BDQN, the agent interacts with the environment through applying the actions proposed by TS, i.e. $a_{\text{TS}}$. We utilize a notion of experience replay buffer where the agent stores its recent experiences. The agent draws $W \sim \mathcal{N}(W^{target}, Cov)$ (abbreviation for sampling of vector $w$ for each action separately) every $T^{sample}$ steps and act optimally with respect to the drawn weights. During the inner loop of the algorithm, we draw a minibatch of data from replay buffer and use loss

$$(y_\tau - \left[W^\top \phi_\theta(x_\tau)\right]_{a_\tau})^2 \tag{8}$$

$$where \quad y_\tau := r_\tau + \left[W^{target \top} \phi_{\theta^{target}}(x_{\tau+1})\right]_{\hat{a}} \, ; \hat{a} = argmax_{a'} \left[W^\top \phi_\theta(x_{\tau+1})\right]_{a'} \tag{9}$$

and update the weights of network: $\theta \leftarrow \theta - \eta \cdot \nabla_\theta (y_\tau - \left[W^\top \phi_\theta(x_\tau)\right]_{a_\tau})^2$.

We update the target network every $T^{target}$ steps and set $\theta^{target}$ to $\theta$. With the period of $T^{Bayes \, target}$ the agent updates its posterior distribution using larger minibatch of data drawn from replay buffer and sample $W$ with respect to the updated posterior. Algorithm 1 gives the full description of BDQN.

# 6 EXPERIMENTS

We apply BDQN on a variety of games in the OpenAiGym [1](Brockman et al., 2016). As a baseline[2], we run DDQN algorithm and evaluate BDQN on the measures of sample complexity and score.

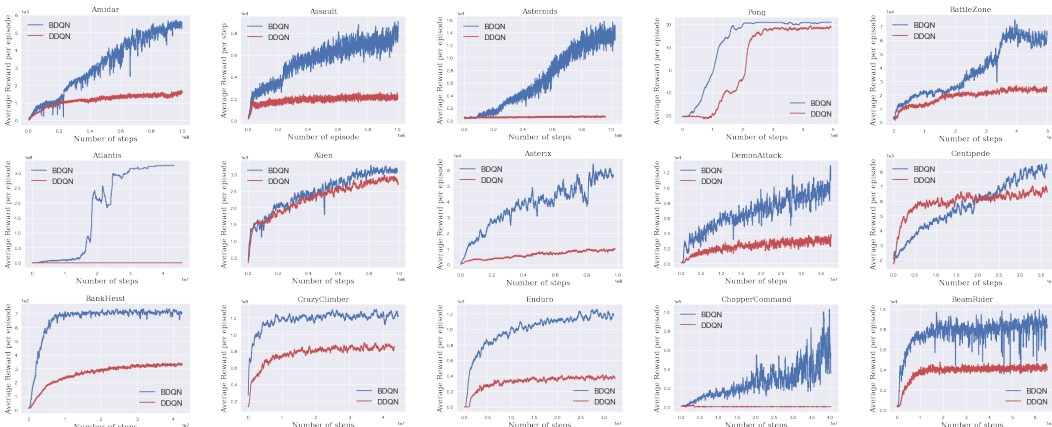

Figure 2: The fast targeted exploration of BDQN

**Network architecture:** The input to the network part of BDQN is $4 \times 84 \times 84$ tensor with a rescaled and averaged over channels of the last four observations. The first convolution layer has $32$ filters of size $8$ with a stride of $4$. The second convolution layer has $64$ filters of size $4$ with stride $2$. The last convolution layer has $64$ filters of size $3$ followed by a fully connected layers with size $512$. We add a BLR layer on top of this.

**Choice of hyper-parameters:** For BDQN, we set the values of $W^{target}$ to the mean of the posterior distribution over the weights of BLR with covariances $Cov$ and draw $W$ from this posterior. For the fixed $W$ and $W^{target}$, we randomly initialize the parameters of network part of BDQN, $\theta$, and train it using RMSProp, with learning rate of $0.0025$, and a momentum of $0.95$, inspired by (Mnih et al., 2015) where the discount factor is $\gamma = 0.99$, the number of steps between target updates $T^{target} = 10k$ steps, and weights $W$ are re-sampled from their posterior distribution every $T^{sample}$ step. We update the network part of BDQN every $4$ steps by uniformly at random sampling a mini-batch of size $32$ samples from the replay buffer. We update the posterior distribution of the weight set $W$ every $T^{Bayes\ target}$ using mini-batch of size $B$ (if size of replay buffer is less than $B$ at the current step, we choose the minimum of these two ), with entries sampled uniformly form replay buffer. The experience replay contains the $1M$ most recent transitions. Further hyper-parameters, are equivalent to ones in DQN setting.

**Hyper-parameters tunning:** For the BLR, we have noise variance $\sigma_\epsilon$, variance of prior over weights $\sigma$, sample size $B$, posterior update period $T^{Bayes\ target}$, and the posterior sampling period $T^{sample}$. To optimize for this set of hyper-parameters we set up a very simple, fast, and cheap hyper-parameter tunning procedure which proves the robustness of BDQN. To fine the first three, we set up a simple hyper-parameter search. We used a pretrained DQN model for the game of *Assault*, and removed the last fully connected layer in order to have access to its already trained feature representation. Then we tried combination of $B = \{T^{target}, 10 \cdot T^{target}\}$, $\sigma = \{1, 0.1, 0.001\}$, and $\sigma_\epsilon = \{1, 10\}$ and test for $1000$ episode of the game. We set these parameters to their best $B = 10 \cdot T^{target}, \sigma = 0.001, \sigma = 1$.

---

[1]Each input frame is a pixel-max of the two consecutive frames. We detailed the environment setting in the implementation code

[2]We also attempted to include Bootstrapped DQNs (Osband et al., 2016) as a baseline. Due to lack of implementation details, we tried difference reasonable way to make it to work but we were not successful to reproduce it despite extensive experimentation. Moreover, they admit in their paper that uniformly chosen DQNs outperforms their setting.

The above hyper-parameter tuning is cheap and fast since it requires only a few times $B$ number of forward passes. For the remaining parameter, we ran BDQN ( with weights randomly initialized) on the same game, *Assault*, for $5M$ time steps, with a set of $T^{Bayes\ target} = \{T^{target}, 10 \cdot T^{target}\}$ and $T^{sample} = \{\frac{T^{target}}{10}, \frac{T^{target}}{100}\}$ where BDQN performed better with choice of $T^{Bayes\ target} = 10 \cdot T^{target}$. For both choices of $T^{sample}$, it performed almost equal where we choose the higher one. We started off with the learning rate of $0.0025$ and did not tune for that. Thanks to the efficient TS exploration and closed form BLR, BDQN can learn a better policy in even shorter period of time. In contrast, it is well known for DQN based methods that changing the learning rate causes a major degradation in the performance, Apx. A. The proposed hyper-parameter search is very simple where the exhaustive hyper-parameter search is likely to provide even better performance. In order to compare the fairness in sample usage, we argue in Apx. A, that the network part of BDQN and its corresponding part in DDQN observe the same number of samples but the BLR part of BDQN uses 16 times less samples compared to its corresponding last layer in DDQN, Apx. A. All the implementations are coded in MXNet framework (Chen et al., 2015) and are available at ..... .

**Results:** The results are provided in Fig. 2 and Table. 2. Mostly the focus of the experiments are on sample complexity in Deep-RL, even though, BDQN provides much larger scores compared to base line. For example, for the game *Atlantis*, DDQN[†] gives score of $64.67k$ after $200M$ samples during evaluation time, while BDQN reaches $3.24M$ after $40M$ samples. As it is been shown in Fig. 2, BDQN saturates for *Atlantis* after 20M samples. We realized that BDQN reaches the internal *OpenAIGym* limit of $max\_episode$, where relaxing it improves score after $15M$ steps to $62M$.

We can observe that BDQN can immediately learn significantly better policies due to its targeted exploration in much shorter period of time. Since BDQN on game *Atlantis* promise a big jump around time step $20M$, we ran it five more times in order to make sure it was not just a coincidence. We did the same additional five experiments for the game *Amidar* as well. We observed that the improvements are consistent among the different runs. For the game Pong, we ran the experiment for a longer period but just plotted the beginning of it in order to observe the difference. For some games we did not run the experiment to $100M$ samples since the reached their plateau.

| Game | BDQN | DDQN | DDQN[†] | Human | SC | SC[†] | Step |
|---|---|---|---|---|---|---|---|
| Amidar | 5.52k | 0.99k | 0.7k | 1.7k | 22.9M | 4.4M | 100M |
| Alien | 3k | 2.9k | 2.9k | 6.9k | - | 36.27M | 100M |
| Assault | 8.84k | 2.23k | 5.02k | 1.5k | 1.6M | 24.3M | 100M |
| Asteroids | 14.1k | 0.56k | 0.93k | 13.1k | 58.2M | 9.7M | 100M |
| Asterix | 58.4k | 11k | 15.15k | 8.5k | 3.6M | 5.7M | 100M |
| BeamRider | 8.7k | 4.2k | 7.6k | 5.8k | 4.0M | 8.1M | 70M |
| BattleZone | 65.2k | 23.2k | 24.7k | 38k | 25.1M | 14.9M | 50M |
| Atlantis | 3.24M | 39.7k | 64.76k | 29.0k | 3.3M | 5.1M | 40M |
| DemonAttack | 11.1k | 3.8k | 9.7k | 3.4k | 2.0M | 19.9M | 40M |
| Centipede | 7.3k | 6.4k | 4.1k | 12.0k | - | 4.2M | 40M |
| BankHeist | 0.72k | 0.34k | 0.72k | 0.72k | 2.1M | 10.1M | 40M |
| CrazyClimber | 124k | 84k | 102k | 35.4k | 0.12M | 2.1M | 40M |
| ChopperCommand | 72.5k | 0.5k | 4.6 k | 9.9k | 4.4M | 2.2M | 40M |
| Enduro | 1.12k | 0.38k | 0.32k | 0.31 | 0.82M | 0.8M | 30M |
| Pong | 21 | 18.82 | 21 | 9.3 | 1.2M | 2.4M | 5M |

Table 2: Comparison of scores and sample complexities(scores in the first two columns are average of 100 consecutive episodes). *SC* represents the number of samples the BDQN requires to bit the human score (Mnih et al., 2015)(" $-$ " means BDQN could not bit) and $SC^{†}$ is the number number of samples the BDQN requires to bit the score of DDQN [†].

## 7 CONCLUSION

In this work we proposed BDQN, a practical TS based RL algorithm which provides targeted exploration in a computationally efficient manner. It involved making simple modifications to the DDQN architecture by replacing the last layer with a Bayesian linear regression. Under the Gaussian prior, we obtained fast closed-form updates for the posterior distribution. We demonstrated significantly faster training and enormously better rewards over a strong baseline DDQN.

This suggests that BDQN can benefit even more from further modifications to DQN such as e.g. Prioritized Experience Replay (Schaul et al., 2015), Dueling approach (Wang et al., 2015), A3C (Mnih et al., 2016), safe exploration (Lipton et al., 2016a), and etc. We plan to explore the benefit of these modifications up to small changes in the future. We also plan to combine strategies that incorporate uncertainty over model parameters with BDQN. In RL, policy gradient (Sutton et al., 2000; Kakade, 2002; Schulman et al., 2015) is another approach which directly learn the policy. In practical policy gradient, even though the optimal policy, given Markovian assumption, needs to be deterministic, the policy regularization is a dominant approach to make the policy stochastic and preserve the exploration thorough the stochasticity of the policy (Neu et al., 2017; Schulman et al., 2017). We plan to explore the advantage of TS based exploration instead of regularizing the policy and make it stochastic.

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

## A  APPENDIX

**Learning rate:**   It is well known that DQN and DDQN are sensitive to the learning rate and change of learning rate can degrade the performance to even worse than random policy. We tried the same learning rate as BDQN, 0.0025, for DDQN and observed that its performance drops. Fig. 3 shows that the DDQN with higher learning rates learns as good as BDQN at the very beginning but it can not maintain the rate of improvement and degrade even worse than the original DDQN.

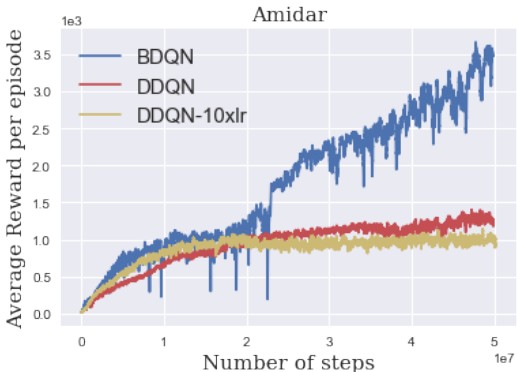

Figure 3: Effect of learning rate on DDQN

**Computational and sample cost comparison:**   For a given period of game time, the number of the backward pass in both BDQN and DQN are the same where for BDQN it is cheaper since it has one layer (the last layer) less than DQN. In the sense of fairness in sample usage, for example in duration of $10 \cdot T^{Bayes\ target} = 100k$, all the layers of both BDQN and DQN, except the last layer, see the same number of samples, but the last layer of BDQN sees 16 times fewer samples compared to the last layer of DQN. The last layer of DQN for a duration of $100k$, observes $25k = 100k/4$ (4 is back prob period) mini batches of size 32, which is $16 \cdot 100k$, where the last layer of BDQN just observes samples size of $B = 100k$. As it is mentioned in Alg. 1, to update the posterior distribution, BDQN draws $B$ samples from the replay buffer and needs to compute the feature vector of them. This step of BDQN gives a superiority to DQN in the sense of speed which is almost $70\%$ faster than BDQN (DQN, on average, for the update does full forward and backward passes while BDQN does not do backward path on the last layer but needs an extra forward pass in order to compute the feature representation). One can easily relax this limitation by parallelizing this step with the main body of BDQN or deploying on-line posterior update methods.

**Thompson sampling frequency:**   The choice of TS update frequency can be crucial from domain to domain. If one chooses $T^{sample}$ too short, then computed gradient for backpropagation of the feature representation is not going to be useful since the gradient get noisier and the loss function is changing too frequently. On the other hand, the network tries to find a feature representation which is suitable for a wide range of different weights of the last layer, results in improper use of model capacity. If the TS update frequency is too low, then it is far from being TS and losses randomized exploration property. The current choice of $T^{sample}$ is suitable for a variety of Atari games since the length of each episode is in range of $\mathcal{O}(T^{sample})$ and is infrequent enough to make the feature representation robust to big changes.

For the RL problems with shorter horizon we suggest to introduce two more parameters, $\tilde{T}^{sample}$ and $\tilde{W}$ where $\tilde{T}^{sample}$, the period that of $\tilde{W}$ is sampled our of posterior, is much smaller than $T^{sample}$ and $\tilde{W}$ is being used just for making TS actions while $W$ is used for backpropagation of feature representation. For game Assault, we tried using $\tilde{T}^{sample}$ and $\tilde{W}$ but did not observe much a difference, and set them to $T^{sample}$ and $W$. But for RL setting with a shorter horizon, we suggest using them.

**Further investigation in Atlantis:**   After removing the maximum episode length limit for the game Atlantis, BDQN gets the score of 62M. This episode is long enough to fill half of the replay buffer and

make the model perfect for the later part of the game but losing the crafted skill for the beginning of the game. We observe in Fig. 4 that after losing the game in a long episode, the agent forgets a bit of its skill and loses few games but wraps up immediately and gets to score of $30M$. To overcome this issue, one can expand the replay buffer size, stochastically store samples in the reply buffer where the later samples get stored with lowers chance, or train new models for the later parts of the episode. There are many possible cures for this interesting observation and while we are comparing against DDQN, we do not want to advance BDQN structure-wise.

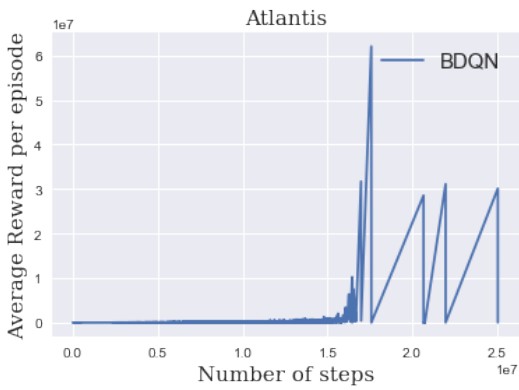

Figure 4: BDQN on Atlantis after removing the limit on max of episode length

