# OpenReview forum: "Efficient Exploration through Bayesian   Deep Q-Networks"
_ICLR.cc/2018/Conference — Reject_

### Official Review · AnonReviewer1 · 2017-11-27
**A nice algorithm. Decent preliminary results, but some more validation would be welcome.**

**Rating:** 6
**Confidence:** 4

**Review:**

The authors propose a new algorithm for exploration in Deep RL. They apply Bayesian linear regression, given the last layer of a DQN network as features, to estimate the Q function for each action. Posterior weights are sampled to select actions during execution (Thompson Sampling style). I generally liked the paper and the approach, here are some more detailed comments.

Unlike traditional regression, here we are not observing noisy realisations of the true target, since the algorithm is bootstrapping on non-stationary targets. It’s not immediately clear what the semantics of this posterior are then. Take for example the case where a particular transition (s,a,r,s’) gets replayed multiple times in a row, the posterior about Q(s,a) might then become overly confident even though no new observation was introduced.

Previous applications of TS to MDPs (Strens, (A Bayesian framework for RL) 2000; Osband 2013) commit to a posterior sample for an episode. But the proposed algorithm samples every T_sample steps, did you find this beneficial to wait longer before resampling? It would be useful to comment on that aspect.

The method is evaluated on 6 Atari games (How were the games selected? Do they have exploration challenges?) against a single baseline (DDQN). DDQN wasn’t proposed as an exploration method so it would be good to justify why this is an appropriate baseline (versus other exploration methods). The authors argue they could not reproduce Osband’s bootstrapped DQN, which is also TS-based, but you could at least have reported their scores.

On these games versus (their implementation of) DDQN, the results seem encouraging. But it would be good to know whether the approach works well across games and is competitive against other stronger baselines. Alternatively, some evidence that interesting exploratory behavior is obtained (in Atari or even smaller domain) would help convince the reader that the approach does what it claims in practice.

In addition, your reported score on Atlantis of ~2M seems too big. Did you cap the max episode time to 30mins? As is done in the baselines usually.


Minor things:
-“TS finds the true Q-function very fast” But that contradicts the previous statements, I think you mean something different. If TS does not select certain actions, the Q-function would not be updated for these actions. It might find the optimal policy quickly though, even though it doesn’t resolve the entire value function completely.
-Which epsilon did you use for evaluation of DDQN in the experiments? It’s a bit suspicious that it doesn’t achieve 20+ in Pong.
-The history of how to go from a Bellman equation to a sample-based update seems a bit distorted. Sample-based RL did not originate in 2008. Also, DQN does not optimize the Bellman residual, it’s a TD update.

---

> ### Author Response · Authors · 2018-01-05
> **Response to AnonReviewer1**
>
> We thank AnonReviewer1 for a clear and constructive review. We are encouraged that you recognize the importance of the problem addressed and the novelty of the methods
>
> Regarding the posterior distribution, we apologize for being imprecise. As we noted to another reviewer, indeed this is not a true posterior but rather an approximation of the Q values that has an explicit smooth (approximate) representation of the uncertainty of the state-action values. We agree it can be off in the described situation but since BDQN runs BLR frequently and uses a moving window of replay buffer it cannot have a severe effect on BDQN performance. Indeed one interesting finding of our work is that this simple approach yields surprisingly large empirical benefits.
>
> Thanks for raising this interesting point. We actually discuss the effect of sampling frequency in the appendix. In the episodic RL, it is enough to Thompson sample the model at the beginning of each episode (theoretically, more frequent sampling does not change the existing bounds). For Atari games we use T^{sample} equal to 1000 which is at the same order of game episode horizon. In the appendix, we added a further discussion about the effect of resampling frequency, the insight about how its design, and what T_sample may be best for the RL problems with shorter or longer horizon.
>
> We would like to encourage the reviewer to look at the latest update of the draft where we added our results on more games and currently, we have 15 games. Unfortunately, due to the high cost of deep RL experiment, we were not able to run bdqn on all the Atari games. Regarding the baseline,
> we choose DDQN as a simple baseline that is quite similar to BDQN except in the last layer and in the fit, and we have clarified this.
> We ran Bootstrap DQN as another baseline for 5 games, but unfortunately, despite extensive experimentation and design choices, we were not able to reproduce their results. For some games, we received the return of less than the return of random policy. For honesty and clarity, since November, we put our Bootstrap DQN implementation public as well.
> Currently, in the TS RL literature, it is one of the biggest challenges in the community to provide a significant improvement, as promised by TS, over DDQN, you can find almost all of the TS based cited literature in our paper, they compare against dqn and ddqn. It is roughly known, also discussed and confirmed with the authors of some of these Thompson sampling works, that neither of the proposed approaches produced much beyond the modest gains of DDQN on Atari games, except, as you correctly point out the current proposed BDQN approach which provides a significant improvement over this baseline.
>
>
> Regarding the game Atlantis, the behavior of bdqn on this game was interesting for us as well.  We elaborate more on the scores of this game in the appendix. To get the score of 3.2M after 20M samples, we enforced the mentioned limit. When we removed this limit, the score gets to 62M after 15M samples.
>
> Regarding finding the optimal Q function and policy, the reviewer is right and we like to thank you also for your point on it. For the described grid world, in order to simplify the example, we assume that the game is deterministic, game horizon H is the episode length as well and only the true Q is the most optimistic function in the set. In this case, any other Q, even those which take the agent to the destination have a non zero Bellman error (defined in https://arxiv.org/abs/1610.09512), therefore the agent wants to eliminate them from the set. We have revised the statement.
>
> We already discussed the experiments in the response to the AC. We ran the mentioned experiment again, and as AnonRev3 confirmed, the current scores are similar to the baseline. Furthermore, in our comparison tables, we compare against the scores of DDQN from its original papers during its evaluation time as well.
>
> We apologize and we have corrected accordingly. We choose the 2008 paper due to its theoretical analysis and have updated accordingly.

---

### Official Review · AnonReviewer2 · 2017-11-29
**Interesting idea but lack of baselines, strong theoretical justification and reference to previous work**

**Rating:** 5
**Confidence:** 4

**Review:**


The authors describe how to use Bayesian neural networks with Thompson sampling
for efficient exploration in q-learning. The Bayesian neural networks are only
Bayesian in the last layer. That is, the authors learn all the previous layers
by finding point estimates. The Bayesian learning of the last layer is then
tractable since it consists of a linear Gaussian model. The resulting method is
called BDQL. The experiments performed show that the proposed approach, after
hyper-parameter tuning, significantly outperforms the epsilon-greedy
exploration approaches such as DDQN.

Quality:

I am very concern about the authors stating on page 1 "we sample from the
posterior on the set of Q-functions". I believe this statement is not correct.
The Bayesian posterior distribution is obtained by combining an assumed
generative model for the data, data sampled from that model and some prior
assumptions. In this paper there is no generative model for the data and the
data obtained is not actually sampled from the model. The data are just targets
obtained by the q-learning rule. This means that the authors are adapting
Q-learning methods so that they look Bayesian, but in no way they are dealing
with a principled posterior distribution over Q-functions. At least this is my
opinion, I would like to encourage the authors to be more precise and show in
the paper what is the exact posterior distribution over Q-functions and show
how they approximate that distribution, taking into account that a posterior
distribution is obtained as $p(theta|D) \propto p(D|theta)p(\theta)$. In the
case addressed in the paper, what is the likelihood $p(D|\theta)$ and what are
the modeling assumptions that explain how $D$ is generated by sampling from a
model parameterized by \theta?

I am also concerned about the hyper-parameter tuning for the baselines. In
section 5 (choice of hyper-parameters) the authors describe a quite exhaustive
hyper-parameter tuning procedure for BDQL. However, they do not mention whether
they perform a similar hyper-parameter tuning for DDQN, in particular for the
parameter epsilon which will determine the amount of exploration. This makes me
wonder if the comparison in table 2 is fair. Especially, because the authors
tune the amount of data from the replay-buffer that is used to update their
posterior distribution. This will have the effect of tuning the width of their
posterior approximation which is directly related to the amount of exploration
performed by Thompson sampling. You can, therefore, conclude that the authors are
tuning the amount of exploration that they perform on each specific problem.
Is that also being done for the baseline DDQN, for example, by tuning epsilon in
each problem?

The authors also report in table 2 the scores obtained for DDQN by Osband et
al. 2016. What is the purpose of including two rows in table 2 with the same
method? It feels a bit that the authors want to hide the fact that they only
compare with a singe epsilon-greedy baseline (DDQN). Epsilon-greedy methods
have already been shown to be less efficient than Bayesian methods with
Thompson sampling for exploration in q learning (Lipton et al. 2016).

The authors do not compare with variational approaches to Bayesian learning
(Lipton et al. 2016). They indicate that since Lipton et al. "do not
investigate the Atari games, we are not able to have their method as an
additional baseline". This statement seems completely unjustified. The authors
should clearly include a description of why Lipton's approach cannot be applied
to the Atari games or include it as a baseline.

The method proposed by the authors is very similar to Lipton's approach. The
only difference is that Lipton et al. use variational inference with a
factorized Gaussian distribution to approximate the posterior on all the
network weights. The authors by contrast, perform exact Bayesian inference, but
only on the last layer of their neural network. It would be very useful to know
whether the exact linear Gaussian model in the last layer proposed by the
authors has advantages with respect to a variational approximation on all the
network weights. If Lipton's method would be expensive to apply to large-scale
settings such as the Atari games, the authors could also compare with that
method in smaller and simpler problems.

The plots in Figure 2 include performance in terms of episodes. However, it
would also be useful to know how much is the extra computational costs of
the proposed method. One could imagine that computing the posterior
approximation in equation 6 has some additional cost. How do BDQN and DDQN
compare when one takes into account running time and not episode count into
account?

Clarity:

The paper is clearly written. However, I found a lack of motivation for the
specific design choices made to obtain equations 9 and 10. What is a_t in
equation 9? The parameters \theta are updated just after equation 10 by
following the gradient of the loss in which the weights of the last layer are
fixed to a posterior sample, instead of the posterior mean. Is this update rule
guaranteed to produce convergence of \theta? I could imagine that at different
times, different posterior samples of the weights will be used to compute the
gradients. Does this create any instability in learning?

I found the paragraph just above section 5 describing the maze-like
deterministic game confusing and not very useful. The authors should improve
this paragraph.

Originality:

The proposed approach in which the weights in the last layer of the neural
network are the only Bayesian ones is not new. The same method was proposed in

Snoek, J., Rippel, O., Swersky, K., Kiros, R., Satish, N., Sundaram, N., ... &
Adams, R. (2015, June). Scalable Bayesian optimization using deep neural
networks. In International Conference on Machine Learning (pp. 2171-2180).

which the authors fail to cite. The use of Thompson sampling for efficient
exploration in deep Q learning is also not new since it has been proposed by
Lipton et al. 2016. The main contribution of the paper is to combine these two
methods (equations 6-10) and evaluate the results in the large-scale setting of
ATARI games, showing that it works in practice.

Significance:

It is hard to determine how significant the work is since the authors only
compare with a single baseline and leave aside previous work on efficient
exploration with Thompson sampling based on variational approximations.

As far as the method is described, I believe it would be impossible to
reproduce their results because of the complexity of the hyper-parameter tuning
performed by the authors. I would encourage the authors to release code that can
directly generate Figure 2 and table 2.

---

> ### Author Response · Authors · 2018-01-05
> **Response to AnonReviewer2**
>
> Thanks for your thoughtful review of our paper. We appreciate it
>
> “Sampling from posterior”: We apologize for being imprecise. As we noted to another reviewer, we have an approximate algorithm in which we use Bayesian linear regression to fit a function to the Q values in a way that allows some uncertainty over the resulting state-action values to be fit, and therefore sampled. We have updated our discussion accordingly. Other algorithms like Bootstrapped DQN also empirically compute an approximation over the posterior of Q values. We believe that our approach gains benefit from two features: (1) from computing an exact linear regression fit to the last layer, which can be more data efficient than single step updates (though in comparing to episodic replay it may vary); and (2) from an explicit (approximate) representation of uncertainty over the Q-values that can be easily sampled and used to inform exploration.
>
> Regarding the hyperparameters, as we also noted in the area chair review, the provided hyperparameter tuning is been used for the tuning of the extra parameters of BDQN and contains a few short run of BDQN. We revised the corresponding section of the paper and made it clear that the performed HPO was simple, quick, and indication of the robustness of BDQN. About per game parameter, we use a fixed size of replay sample for BLR among all the games.
>
> We apologize if the tables have appeared misleading. We were merely trying to illustrate the performance of BDQN vs DDQN after the same number of samples, and in addition, the BDQN in its earlier stage vs DDQN after 200M samples during its evaluation time (reduced epsilon) which is reported at the original paper. We also have been asked and advised to add the comparison to human score, samples complexity of reaching the human scores and sample complexity of reaching DDQN^\dagger score.
>
>
> Regarding BBQ work, as we mentioned above, we believe that our approach gains benefit from two features: (1) from computing an exact linear regression fit to the last layer and (2) from an explicit (approximate) representation of uncertainty over the Q-values. In the TS deep RL literature, mostly, two lines of works for Thompson sampling have been studied. One is variational inference based method,e.g.
> Efficient Dialogue Policy Learning with BBQ-Networks
> Dropout as a Bayesian Approximation
>
> And the second one is the confidence based, e.g.
> Deep exploration via randomized value functions
> Deep exploration via bootstrapped dqn
> The uncertainty bellman equation and exploration
> Etc.
> Where the objective functions in these two lines of works, as the reviewer also mentioned, are different and BDQN is in the second category. In addition, these two following papers:
> Deep exploration via bootstrapped dqn
> Dropout Inference in Bayesian Neural Networks with Alpha-divergences
> Argue that how the variational inference methods can severely underestimate model uncertainty.
>
> We appreciate the reviewer for mentioning the typo in our figure2. The figure 2 is updated and the episode count was a typo, it is the step count. We compare the computation cost of BDQN  and DDQN, but computing equation 6 just involves inverting a matrix of 512 by 512, every 100k time step, which is computationally negligible. We have a detailed discussion on the computation cost in the appendix.
>
> a_\tau is the action taken at time step \tau, we restated it in the main text.
>
> Regarding the use of W in the update of theta, we are grateful to the reviewer for the careful review of our paper. The concern in use of W for feature representation update is really a keen observation and we elaborated it in the appendix. As the reviewer mentioned, we should not change this W too frequently, since it forces the network to spend the network capacity for providing a feature representation which is good for the variety of different draws of W. At the same time, it provides a noisy gradient since the cost surface changes fast and prevent the model from learning. On the other hand, the update of W, used for the update of theta, should not happen barely as well since every 100K the posterior gets updated and the feature representation should not overfit to a single W, and consequently overfit to a fixed set of skills.
>
> We thank the reviewer for the feedback on the clarity of section 5, we apologize and we have updated this section.
>
> We added Snoek et al to our paper. Thanks for mentioning it.
>
> We agree that reproducibility is critical and we had released our code including hyperparameter settings in November. Also, the graphs, learned models (model parameters), returns per episode, any remaining output of the experiments and the required material for reproducing the plots are provided publicly.

---

### Official Review · AnonReviewer3 · 2017-12-04
**Interesting approach for Thompson sampling in DQN, some concerns over baseline.**

**Rating:** 5
**Confidence:** 4

**Review:**

(Last minute reviewer brought in as a replacement).

This paper proposed "Bayesian Deep Q-Network" as an approach for exploration via Thompson sampling in deep RL.
This algorithm maintains a Bayesian posterior over the last layer of the neural network and uses that as an approximate measure of uncertainty.
The agent then samples from this posterior for an approximate Thompson sampling.
Experimental results show that this outperforms an epsilon-greedy baseline.

There are several things to like about this paper:
- The problem of efficient exploration with deep RL is important and under-served by practical algorithms. This seems like a good algorithm in many ways.
- The paper is mostly clear and well written.
- The experimental results are impressive in their outperformance.

However, there are also some issues, many of which have already been raised:
- The poor performance of the DDQN baseline is concerning and does not seem to match the behavior of prior work (see Pong for example).
- There are some loose and misleading descriptions of the algorithm computing "the posterior" when actually this is very much an approximation method... that's OK to have approximations but it shouldn't be hidden away.
- The connection to RLSVI is definitely understated, since with a linear architecture this is precisely RLSVI. The sentiment that extending TS to larger spaces hasn't been fully explored is definitely valid... but this line of work should certainly be mentioned in the 4th paragraph. RLSVI is provably-efficient with a state-of-the-art regret bound for tabular learning - you would probably strengthen the case for this algorithm as an extension of RLSVI by building on this connection... otherwise it's a bit adhoc to justify this approximation method.
- This paper spends a lot of time re-deriving Bayesian linear regression in a really standard way... and without much discussion of how/why this method is an approximation (it is) especially when used with deep nets.

Overall, I like this paper and the approach of extending TS-style algorithms to Deep RL by just taking the final layer of the neural network.
However, it also feels like there are some issues with the baselines + being a bit more clear about the approximations / position relative to other algorithms for approximate TS would be a better approach.
For example, in linear networks this is the same as RLSVI, bootstrapped DQN is one way to extend this idea to deep nets, but this is another one and it is much better because XYZ. (this discussion could perhaps replace the rather mundane discussion of BLR, for example).

In it's current state I'd say marginally above, but wouldn't be surprised if these changes turned it into an even better paper quite quickly.


===============================================================

Revising my review following the rebuttal period and also the (ongoing) revisions to the paper.

I've been disappointed by the authors have incorporated the feedback/reviews - I expected something a little more clear / honest. Given the ongoing review decisions/issues I'm putting my review slightly below accept.

## Relation to literature on "randomized value functions"
It's really wrong to present BDQN as is if it's the first attempt at large-scale approximations to Thompson sampling (and then slip in a citation to RLSVI as a BDQN-like algorithm). This algorithm is a form of RLSVI (2014) where you only consider uncertainty over the last (linear) layer - I think you should present it like this. Similarly *some* of the results for Bootstrapped DQN (2016) on Atari are presented without bootstrapping (pure ensemble) but this is very far from an essential part of the algorithm! If you say something like "they did not estimate a true posterior" then you should quantify this and (presumably) justify the implication that taking a gaussian approximation to the final layer is a *true* posterior. In a similar vein, you should be clear about the connections to Lipton et al 2016 as another method for approximate Bayesian posteriors in DQN.

## Quality/science of experiments
The experimental results have been updated, and the performance of the baseline now seems much more reasonable. However, the procedure for "selecting arbitrary number of frames" to report performance seems really unnecessary... it would be clear that BDQN is outperforming DDQN... you should run them all for the same number of frames and then either compare (final score, cumulative score, #frames to human) or something else more fair/scientific. This type of stuff smells like overfitting!

---

> ### Author Response · Authors · 2018-01-05
> **Response to AnonReviewer3**
>
> Thank you very much for your careful and constructive feedbacks, they helped us to carve the face of our paper in a strong way. We are glad that you appreciated both the method and the clarity of exposition.
>
> We completely agree with your comments with respect to RLSVI and Bootstrapped DQN,  we have revised the discussion accordingly and presented bdqn as a direct extension of RLSVI.
>
> You are right about our approach is also an approximate method of a posterior. We apologize for being imprecise and we have updated our language to be more careful in our discussion and presentation of Bayesian linear regression.
>
> Regarding your point about the number of frames to run, for the game pong, we have run the experiments for a longer period of time, but in order to observe the difference between bdqn and ddqn performances, we had to plot just for the beginning of the game. For some games, we stopped when the plots reached the area around the plateau and for some games, we ran out resources to continue to 100M.
> We would like to share with the reviewer that all the returns per episode, frame counts, and clipped reward return arrays are shared publicly and one can easily reproduce these plots and compare the score in different time steps. We also added a few more columns for the sample complexity and comparison to human scores.
>
> Regarding “running for the same number of samples”, we should mention that in the reported tables, the scores in the first column (BDQN) and the second column (DDQN) are the scores after the same number of samples (provided in the last column). For each game, both bdqn and ddqn are run for the same number of samples. The DDQN^\dagger is also the score of DDQN reported in the original DDQN paper during the evaluation phase.
>
> Regarding the baseline, as you mentioned, the current plots and scores are updated.
>
> Regarding re-deviation of BLR, we tried to balance between deriving the standard method while making the paper self-contained. We received feedback that it is helpful to have BLR derivation, e.g. Rev2  suggested to elaborate more on BLR part.
>
>
> Again thanks to your feedback, we made another constructive change, with help of other TS researchers, in the presentation of our paper and overview of related work.
> We start off with PSRL, and how randomised value function and  RLSVI leveraged it. Then how bootstrap dqn extend the idea to deep learning, followed by the noisy net, bbq, shallow UBE and LS-DQN. Finally, we explain that bdqn is an extension to RLSVI and follows the same idea.
>
> Still, we would be happy to get more feedback from you to further improve our paper.

---

### Public Comment · (anonymous) · 2017-11-27
**Question marks on the baseline**

Your baseline results of DDQN seem strange to me, particularly the result on Pong.
It seems like these results are quite different from (for example) https://github.com/openai/baselines

Can you comment on this?

---

> ### Author Response · Authors · 2017-11-28
> **Baseline**
>
> Thanks for the comment. For the baseline, we used the implementation described in DDQN paper ‘’Deep Reinforcement Learning with Double Q-learning’’, and the available code in
> https://github.com/kazizzad/Double-DQN-MxNet-Gluon.git
> Since there's always also a variance across runs, we ran the code again, where we got a score close to what mentioned in DDQN paper. We'll update the plot for the revision. Thanks for mentioning it.
>
> Cheers,
> Authors

---

### Public Comment · ~Ian_Osband1 · 2017-11-27
**Connection to RLSVI**

Cool work!

I wanted to highlight a deeper connection between your work and the algorithm RLSVI that you already cite, but maybe didn't realize the deeper connection: https://arxiv.org/pdf/1402.0635.pdf.

If you run BDQN with a linear architecture and T^{sample} = H finite horizon problem, my understanding is that BDQN is exactly the same as RLSVI. Certainly RLSVI is presented in that paper for a linear architecture, but the general approach of Randomized Least Squares Value Iteration is not specific to that architecture https://searchworks.stanford.edu/view/11891201.

It is very interesting though that you get better performance using this "last-layer" approach to RLSVI, rather than something like Bootstrap/Ensemble. Maybe one way to present this is as an effective way to extend RLSVI to multi-layer architectures.

By the way, I find it surprising that resampling the noisy W in this way so infrequently is not simply learned away by the SGD... can you comment on this?

---

> ### Author Response · Authors · 2017-11-28
> **RLSVI**
>
> Hi Ian
> Thank you for your comment. We are aware of your work on RLSVI, and we agree that if we make the BDQN episodic and do not update the feature representation it is exactly RLSVI. We will elaborate it more on the main draft.
>
> Regarding using RLSVI or Bayesian regression at each layer, that is an interesting extension to BDQN and we have left it for the future work.
>
> For the revision, we are adding a further plot to the game Atlantis. In DDQN paper ‘’Deep Reinforcement Learning with Double Q-learning’’, the score for this game is 65k and you can notice that, Fig. 2, the BDQN agent suddenly starts to learn a significantly better policy which gives an average score of 3.2M, then stays there, with no improvement. We investigated this stopping in the improvement by looking at the episode length. We realized that the agent reaches the maximum episode length limit of openaigym which is 100k. After removing this limit, surprisingly it got a score of 62M after 15M samples which is almost 1000x higher than the reported one in DDQN paper.
>
> About resampling from posterior of W, actually in the older version of the algorithm, there was another parameter \tilde{W} which was sampled from the same distribution as W, but more frequently (every \tilde{T} time step). We used \tilde{W} to make decisions and used W (samples every T^{sample}) in order to update the feature representation in Bellman residual equation (line 14 in the Alg). We tried \tilde{T} of 1, 10, 100, and  T^{sample} time steps for game of Assault during the hyperparameter tuning period, but did not observe any significant difference. That’s why we, for simplicity, removed it from the setting and just kept W. It looks for Atari games, sampling more frequency does not make much difference (T^{sample} is in the same (or less) order as episode length (H) for many Atari games), but for the RL problems with shorter horizon and especially deterministic transition, we believe it makes a difference. We are adding a further discussion in the appendix about sampling frequency of \tilde{W} and address how crucial the choice of \tilde{T} could be in different RL settings.
>
> Cheers
> Authors
>
> “To preserve double-blind status, we won't post the GitHub link here.”

---

### Public Comment · (anonymous) · 2017-12-08
**Related works in the literature**

I think this is a very nice paper. I wanted to ask you about an interesting connection between this work and another recent work on RL exploring (https://arxiv.org/pdf/1709.05380.pdf), in particular the use of the last layer NN statistics to generate uncertainty combined with Thompson sampling. Can you comment on the differences and similarities between your work and this? Thanks.

---

> ### Author Response · Authors · 2017-12-17
> **Different source of uncertainties**
>
> Thanks for your interest in our paper!
> The mentioned paper is an interesting line of work on uncertainty measure for exploration but the source of uncertainty in this paper is different from BDQN. As described in both papers, if one approximates Q(x , a), the mean of random return in state x after taking action a, as a linear function of features
>
> Q(x,a) = w*\phi
>
> then the random return is distributed as follows
>
> Q(x,a)+\epsilon = w^\top\phi+epsilon
>
> From a frequentist perspective, given the data, one can minimize bellman residual (a mean square error in this setting), find a fixed point of it and estimate w^*. Due to the noise \epsilon in the random return and the approximated generative model, the estimated w^* has a frequentist uncertainty and the authors use this uncertainty to randomize over the actions. As it is well known in the linear regression setting, if the noise gets big, the confidence interval gets big as well.
>
> On the other hand, in our setting, the source of uncertainty comes from the agent posterior belief on the Q function and the Thompson sampling is applied over approximated posterior distribution. We approximate the generative model with the Bayesian approach where the parameter w is assigned a prior. The uncertainty on the Q function comes from the posterior belief of w (Eq6) where the randomness in the return is captured by Eq7. Our uncertainty comes from belief in w which is constructed from both randomnesses in return and the prior belief in w.
>
> To be more abstract, the mentioned paper exploits the frequentist uncertainty while in our setting we exploit Bayesian belief to construct the uncertainty.
>
> Cheers,
> Authors

---

> > ### Comment · AnonReviewer3 · 2017-12-18
> > **This doesn't feel like a convincing answer to me**
> >
> > Both of these algorithms use a linear approximation to the final layer of a neural network and the covariance matrix (X^T X) to quantify confidence sets.
> >
> > The question of whether you term this "Bayesian" or "Frequentist" uncertainty feels misleading and a little pretentious. In both cases they are being used to guide an agent's exploration, the epistemological roots of Bayes/Frequency don't seem like the pressing issue.
> >
> > The relevant issue is how to propagate uncertainty over the value function over multiple time steps. One does this via a sampling procedure and the other by attempting to "learn" an approximating neural network from "shallow" = one step uncertainty. This would be good to discuss in the paper.

---

> > > ### Author Response · Authors · 2017-12-19
> > > **Detailed uncertainty approximation**
> > >
> > > Thank you for helping us in addressing this comment. We agree that adding a further discussion in a detailed comparison of UBE and BDQN makes the current draft stronger.
> > >
> > > As the reviewer3 mentioned, UBE provides a Bellman-like equation for uncertainty and learns a shallow network in order to approximate the uncertainty. While in BDQN, the uncertainty is approximated using BLR.

---

### Comment · Area_Chair · 2017-12-11
**Lack of empirical rigor?**

All the reviews seem to question the empirical rigor of this work.  AnonReviewer3 commented that the implemented baseline DDQN didn't seem to match prior work.  AnonReviewer2 also had concerns that the hyperparameter tuning of the baseline DDQN was weaker than their method.  Similarly AnonReviewer1 asked for stronger validation and stronger baselines and points out e.g. "It’s a bit suspicious that it doesn’t achieve 20+ in Pong".

In light of recent revelations in deep reinforcement learning (i.e. https://arxiv.org/pdf/1709.06560.pdf), this seems like a significant issue that is prevalent.  Could the reviewers and authors comment about whether they feel that the empirical evidence presented in this work is strong enough to justify that this paper should not be subject to the criticism presented in the aforementioned paper?

---

> ### Author Response · Authors · 2018-01-05
> **Empirical scores of DDQN**
>
> Area Chair:
> Thanks for the review! We really appreciate it.
>
> Regarding the score of DDQN, we ran the experiments, e.g. for the game Pong, again and reported the result in the current draft. As the AnonReviewer1 also mentioned after visiting the current draft, the current results are similar to the original paper. We are grateful to all the reviewers for their constructive suggestion and we believe that it made the paper stronger.
>
> It would be helpful to mention that, on the score tables in the main draft, we also compared against the score of DDQN^\dagger, which is the reported score of DDQN from the original DDQN paper (copied) during the evaluation phase where evaluation epsilon is set to 0.001 after 200M samples. For BDQN, we did not design any evaluation phase. We already elaborated it more in the main text in order to make it more clear.
>
>
> For the choice of hyper parameters, we used the hyper parameters used in DQN and DDQN, which are tuned through an exhaustive hyperparameter tuning procedure in the original papers for these algorithms.
>
> Regarding the hyperparameter tuning of extra parameters of BDQN, one of the main reason why we talked about it in the main text, aside from being honest about BDQN, was to deliver the point that it is not exhaustive and it is a simple tuning procedure as another proof of the superiority of Thompson sampling over epsilon-greedy. The whole process of hyperparameter tuning, including coding, contains a few runs of BDQN, each for a few hours. We stated that a further hyperparameter tuning can be done for BDQN to provide even more exciting results. We, already, discussed it in the main draft.
>
> The BDQN code is available to the public and is online since November. To preserve double-blind status, we won't post the GitHub link here but it's not too hard to find.

---

### Comment · AnonReviewer3 · 2018-01-03
**[Late] another paper to discuss / cite**

"Shallow Updates for Deep Reinforcmeent Learning"
https://arxiv.org/pdf/1705.07461.pdf

This work on LS-DQN follows a (relatively) similar narrative of using a Least-Squares RL on the last layer of DQN.
You might consider that this paper on Bayesian DQN is similar to an RLSVI-version of LS-DQN.

I was not aware of this until recently.

---

> ### Author Response · Authors · 2018-01-05
> **LS-DQN**
>
> Thanks for the thoughtful review, follow-up of our paper and thank you for mentioning this interesting and related work. This work, which just came out recently and appeared at NIPS, is interestingly very similar to our last layer regression narrative and we already included a discussion about it in our paper.

---

### Author Response · Authors · 2018-01-05
**General reply to reviewers and area chair**

Dear reviewers and area chair

We would like to thank the reviewers and area chair for their thoughtful responses to our paper. We are grateful to each of you for critical suggestions that helped us to significantly improve our paper. Please find individual replies to each of the reviews in the respective threads.

Furthermore, since the Area Chair provided the abstract of the main concerns in the reviews, we would like to ask all the reviewers to consider looking at the area chair’s review and our reply.

---

### Decision · Program_Chairs · 2018-01-29
**ICLR 2018 Conference Acceptance Decision**

**Decision:**

Reject

**Comment:**

This work develops a methodology for exploration in deep Q-learning through Thompson sampling to learn to play Atari games.  The major innovation is to perform a Bayesian linear regression on the last layer of the deep neural network mapping from frames to Q-values.  This Bayesian linear regression allows for efficiently drawing (approximate) samples from the network.  A careful methodology is presented that achieves impressive results on a subset of Atari games.

The initial reviews all indicated that the results were impressive but questioned the rigor of the empirical analysis and the implementation of the baselines.  The authors have since improved the baselines and demonstrated impressive results across more games but questions over the empirical analysis remain (by AnonReviewer3 for instance) and the results still span only a small subset of the Atari suite.  The reviewers took issue with the treatment of related work, placing the contributions of this paper in relation to previous literature.

In general, this paper shows tremendous promise, but is just below borderline.  It is very close to a strong and impressive paper, but requires more careful empirical work and a better treatment of related work.  Hopefully the reviews and the discussion process will help make the paper much stronger for a future submission.

Pros:
- Very impressive results on a subset of Atari games
- A simple and elegant solution to achieving approximate samples from the Q-network
- The paper is well written and the methodology is clearly explained

Cons:
- Questions remain about the rigor of the empirical analysis (comparison to baselines)
- Requires more thoughtful comparison in the manuscript to related literature
- The theoretical justification for the proposed methods is not strong